# Is Pulmonary Involvement a Distinct Phenotype of Post-COVID-19?

**DOI:** 10.3390/biomedicines11102694

**Published:** 2023-10-02

**Authors:** Krystian T. Bartczak, Joanna Miłkowska-Dymanowska, Małgorzata Pietrusińska, Anna Kumor-Kisielewska, Adam Stańczyk, Sebastian Majewski, Wojciech J. Piotrowski, Cezary Lipiński, Sebastian Wawrocki, Adam J. Białas

**Affiliations:** 1Department of Pneumology, Medical University of Lodz, 90-153 Lodz, Poland; joanna.milkowska-dymanowska@umed.lodz.pl (J.M.-D.); anna.kumor-kisielewska@umed.lodz.pl (A.K.-K.); sebastian.majewski@umed.lodz.pl (S.M.); wojciech.piotrowski@umed.lodz.pl (W.J.P.); adam.bialas@umed.lodz.pl (A.J.B.); 2Department of Clinical Pharmacology, Medical University of Lodz, 90-153 Lodz, Poland; adam.stanczyk@umed.lodz.pl; 3The Center for Innovation and Technology Transfer, Medical University of Lodz, 92-215 Lodz, Polandsebastian.wawrocki@siaf.uzh.ch (S.W.); 4Swiss Institute of Allergy and Asthma Research (SIAF), University of Zurich, 7265 Davos, Switzerland; 5Department of Pulmonary Rehabilitation, Center for Lung Diseases and Rehabilitation, Blessed Rafal Chylinski Memorial Hospital for Lung Diseases, 91-520 Lodz, Poland

**Keywords:** post-COVID-19 condition, post-COVID-19 recovery, pulmonary impairment, lung involvement, pandemic

## Abstract

(1) Background: COVID-19 infection often provokes symptoms lasting many months: most commonly fatigue, dyspnea, myalgia and mental distress symptoms. In this study, we searched for clinical features of post-COVID-19 condition (PCC) and differences between patients with and without pulmonary involvement. (2) Methods: A total of 282 patients with a mean age of 57 years (SD +/− 12 years) underwent assessment up to 12 weeks after COVID-19 recovery. The course of acute disease, past medical history and clinical symptoms were gathered; pulmonary function tests were performed; radiographic studies were assessed and follow-up examinations were conducted. Patients with and without detectable pulmonary lesions were divided into separate groups. (3) Results: Patients within the pulmonary group were more often older (59 vs. 51 y.o.; *p* < 0.001) males (*p* = 0.002) that underwent COVID-19-related hospitalization (*p* < 0.001) and were either ex- or active smokers with the median of 20 pack-years. We also managed to find correlations with hypertension (*p* = 0.01), liver failure (*p* = 0.03), clinical symptoms such as dyspnea (*p* < 0.001), myalgia (*p* = 0.04), headache (*p* = 0.009), sleeplessness (*p* = 0.046), pulmonary function tests (such as FVC, TLCO, RV and TLC; *p* < 0.001) and several basic laboratory tests (D-dimer, cardiac troponin, WBC, creatinine and others). (4) Conclusions: Our results indicate that initial pulmonary involvement alters the PCC, and it can be used to individualize clinical approaches.

## 1. Introduction

Since 2020, the scientific world has been observing and acting inside a continuously evolving pandemic caused by severe acute respiratory syndrome coronavirus 2 (SARS-CoV-2). Thousands of research studies regarding acute phase coronavirus disease 2019 (COVID-19) allowed the natural history and risk factors to be described [1] as well as provided clinicians with instruments for the effective management of this viral pneumonia and its short-term consequences [2].

As the convalescent numbers constantly rise, the number of patients with post-COVID-19 health complications observed weeks and months after infection simultaneously increases. A cohort study from Stockholm [3], which included 204,805 participants, assessed the proportion of patients receiving a post-COVID-19 condition (PCC) diagnosis during follow-up to be 1.5% in the whole group; 1% among non-hospitalized, 6% among the hospitalized, and 32% among the ICU-treated individuals. Fatigue, dyspnea, mental distress symptoms (such as sleep disorder or memory problems), and myalgia were reported most often in existing data regardless of the time points being 3, 6, 12, or even 24 months after the acute episode [4,5,6]. Big-group meta-regression of more than 1.2 million patients confirms these chief manifestations and suggests a more frequent association with female sex, age over 20, and hospitalization [7,8].

PCC is defined as the continuation or development of new symptoms after 3 months from infection, lasting for a minimum period of another 2 months [9]. Its manifestations derive from multiple mechanisms that the virus has the potential of inflicting [10]; SARS-CoV-2 interacts with the host microbiome/virome, paving the way for the reactivation of pathogens such as herpesviruses or fungi under the conditions of immune dysregulation [11,12,13,14,15]; it damages the vessel endothelium and provokes clotting issues [16,17]; it alters the brainstem and vagus nerve signaling because of its neuroinvasive capabilities [18,19,20]; finally, it provokes autoimmunity due to immune cells’ overactivity and molecular mimicry between pathogens and the host proteins [21,22].

Therefore, persisting symptoms should be observed because of prolonged injury to specific organs and/or its SARS-CoV-2 preservation in certain tissues [23]. In our study, we pursued the clinical features of PCC, paying special attention to potential differences that survivors with and without detectable lung involvement would present during their slow recovery course.

## 2. Materials and Methods

This prospective observational study was performed in the Outpatient Pneumology Clinic. It has been approved by the Local Bioethical Committee (Approval No. RNN/317/20/KE, approved on 15 December 2020). All research was performed under relevant guidelines and regulations, including the Declaration of Helsinki. Patients signed an informed consent to participate and became involved in the study during their first outpatient visit, taking place up to 12 weeks after COVID-19 recovery. Participants were infected and diagnosed with COVID-19 between June 2020 and April 2021 during the second COVID-19 wave in Poland with the predominant original Wuhan virus variant. The only exclusion criterion was the lack of patient agreement to participate in the study. We experienced much confusion about naming the post-COVID-19 condition (PCC), long-COVID and post-acute COVID-19 syndrome (PACS) in the existing literature, so we decided to continue with unified post-COVID-19 condition (PCC) nomenclature in the following considerations. Quoting the National Institute for Health and Care Excellence (NICE), we define PCC as “signs and symptoms that develop during or after an infection consistent with COVID-19, continue for more than 12 weeks and are not explained by an alternative diagnosis. Post-COVID-19 syndrome may be considered before 12 weeks while the possibility of an alternative underlying disease is also being assessed” [24].

During the assessment, clinical data such as the intensity of cough and dyspnea during acute COVID-19 and post-COVID-19 phases, dyspnea, headache, chest pain, muscle pain, diarrhea, fever, taste and smell impairment, skin lesions, mood worsening, mobility, fatigue assessed by standardized fatigue assessment score (FAS) and body mass index (BMI) score were gathered. Blood samples were drawn, and basic analyses were made, including full blood count with white cell differentiation, aminotransferases, bilirubin, D-dimer, CRP, troponin T and creatinine. Pulmonary function tests (PFTs), including spirometry, lung transfer for carbon monoxide (T_L,CO_) and the 6-min walk test (6 MWT), were performed in most of the patients alongside physical examination. Every patient’s radiological examinations were assessed in an initial and follow-up radiographic study. The group was further divided for analysis into two subgroups based on the initial radiographic study: with and without lung involvement (pulmonary and non-pulmonary post-COVID-19, respectively). Scans from both acute COVID-19 and post-COVID-19 were evaluated individually by two board-certified experts with a minimum of 5 years of experience in lung tomography assessment. Duration of hospitalization and time from confirmed diagnosis of SARS-CoV-2 infection to the assessment was noted. Lung involvement was assessed on a 5-point scale: 0: none (no lung involvement, the non-pulmonary group); 1: 1–25% of lung surface; 2: 26–50% of lung surface; 3: 51–75% of lung surface, and 4: 76–100% of lung surface.

Statistical analysis was performed using R software for MacOS. Continuous data were presented as the mean with SD or median with interquartile range (IQR), depending on the distribution of data. Variables were compared using the unpaired Student’s *t*-test, Welch *t*-test or the Wilcoxon rank-sum test with continuity correction, depending on data normality and homogeneity of variance. Categorical data were analyzed by Pearson’s Chi-squared test or Fisher’s Exact Test according to the tests’ assumptions. Furthermore, multiple logistic regression analysis with stepwise model selection was used to create models to predict which factors were associated with a fibrotic lung pattern. The models were further evaluated by area under the receiving operating characteristic curve (AUROC) analysis [25]. Missing data were not imputed in the analysis.

## 3. Results

A total of 282 patients with a mean age of 57 years (SD +/− 12 years) and BMI of 28.01 (IQR 25.64–31.61) were involved in the study. The median time of first outpatient assessment was 81 days from COVID-19 diagnosis (IQR 59–107.5). Participants’ characteristics are presented in Table 1. Subjects were divided into two groups (further called “phenotypes”) according to the presence of pulmonary lesions in the first radiological examination—a non-pulmonary (no radiological findings) and a pulmonary (presence of lesions) group, comprising 78 (27.66%) and 204 (72.34%) individuals, respectively (Figure 1).

Our findings suggest the presence of significant differences among patients with and without pulmonary involvement (Table 2). The pulmonary phenotype was significantly more often associated with older age (59 vs. 51 years; *p* < 0.001); symptoms such as dyspnea during an acute COVID-19 phase (*p* < 0.001) and dyspnea during PCC visit (*p* < 0.001), duration of myalgia (*p* = 0.04), headache (*p* = 0.009) and sleeplessness (*p* = 0.046) occurrence. These participants were more often hospitalized (*p* < 0.0001), males (*p* = 0.002), with lower spontaneous SpO2 (*p* = 0.03), and requiring a longer hospital stay (*p* = 0.03) with the administration of oxygen (*p* = 0.03; 114/282; 40,43%) and steroid (*p* = 0.0006) therapy. From the whole cohort, 114 patients were treated with oxygen. Only four patients from the whole cohort survived the critical phase of respiratory failure, two of whom required invasive and another two of whom required non-invasive ventilatory support during their hospital stay.

Regarding pre-existing burdens, patients with pulmonary manifestations more commonly reported hypertension (*p* = 0.01) or liver failure (*p* = 0.03; seven patients in the whole group studied). The smoking status itself was not significant, but the cumulated pack-years (*p* = 0.04) made the difference. Overall, 38 participants reported a previous diagnosis of obstructive pulmonary disease (13.48%), and 1 had recognized obstructive sleep apnea (0.004%) with no predilection to the pulmonary group. On the other hand, the history of myocardial infarction (*p* = 0.07) before COVID-19 and skin lesions during the acute phase (*p* = 0.09) were close to reaching significance.

Logically, the most significant alterations between our study groups were the results of PFTs: T_L,CO_ (%; *p* = 0.02; Figure 2), RV (%; *p* < 0.001), TLC (%; *p* < 0.001), and FVC (*p* = 0.04); FEV_1_ alterations almost reached significance (*p* = 0.052). Surprisingly, simple laboratory parameters varied as well: troponin (*p* = 0.003), D-dimer (*p* = 0.004), creatinine (*p* = 0.004), WBC (*p* = 0.03), monocyte (*p* = 0.002) and immature granulocyte count (*p* = 0.002), along with medium red cell volume and size—RDW-CV (*p* = 0.004), RDW-SD (*p* = 0.008).

In the pulmonary group, the median CT score was 2 (26–50% of lung surface; IQR 1–3); in the non-pulmonary group, the CT score remained 0 (no detectable lung involvement, 0% of lung surface; IQR 0–0). Therefore, persistent pulmonary lesions remained specific for the pulmonary cohort only. Regarding the wide range of radiographic abnormalities, usually mixed in their appearance (including regions of ground glass, crazy paving, bronchiectasis, rarely definite fibrosis), we did not seek further conclusions based on radiological studies. We found no difference in the body mass index (BMI), heart rate and blood pressure; duration of fatigue (including FAS score) and fever; FAS score; initial lung involvement and time from diagnosis to follow-up assessment.

Additionally, we performed further analysis of the pulmonary subgroup, and we managed to identify that the extent of radiological lesions significantly correlates with symptoms such as the length of myalgia (*p* = 0.03) and length of hospital stay (*p* = 0.03) as well as dyspnea during acute COVID-19 phase (*p* = 0.007); vital parameter measurements: heart rate (*p* = 0.006), systolic (*p* = 0.03) and diastolic (*p* = 0.02) blood pressure; SpO2 (*p* = 0.03); PFT results: FVC (%; *p* < 0.0001), RV (*p* = 0.0004), TLC (absolute; *p* < 0.0001), and T_L,CO_ (*p* < 0.0001); and laboratory parameters: D-dimer (*p* = 0.004), monocyte level (*p* = 0.02), and RDW-CV (*p* = 0.01). There were no correlations with patients’ age, the time from diagnosis, current dyspnea intensity (*p* = 0.06), cough intensity during both acute COVID-19 and post-COVID-19 phases, 6-min walking test distance, and hemoglobin level.

After stepwise selection, the logistic regression model revealed that the variables regarding age (*p* < 0.001), male sex (*p* < 0.001), frequency of low T_L,CO_ (*p* = 0.03), and severity of dyspnea during active COVID-19 (*p* < 0.001) were significantly associated with the pulmonary phenotype of post-COVID-19 syndrome (Table 3). AUROC analyses showed very good fitness of the model to the data (Table 3, Figure 3).

## 4. Discussion

Nowadays, much is known about the average recovery after COVID-19 pneumonia [26], persistent symptoms, and the level of nuisance they cause. Sequelae consist of constitutional, cardiopulmonary, neurological, and gastrointestinal symptoms that are altogether called the post-COVID-19 condition. Although much, in general, is known about the correlation between COVID-19 severity and PCC [27], the data mostly cover patients with the most frequent organ involvement—meaning the lungs—and there is little information about patients without respiratory failure and lung involvement. This study aimed to distinguish the clinical characteristics of patients with radiologically detectable lung involvement and those with other dominant presentations of SARS-CoV-2 infection (and zero pulmonary lesions described). We managed to show several differences between those groups, while the pulmonary involvement was significantly more often associated with male sex, dyspnea, headache, myalgia and sleeplessness, previously recognized hypertension, liver failure and cumulated pack-years. Logically, these patients were more often hospitalized and for a longer period of time, requiring oxygen and pharmacological treatment as well as presenting worse outcomes on pulmonary function tests. We also managed to find associations with several of the laboratory parameters mentioned in the other paragraphs.

There is much data about PCC from studies on a general population, but vast analyses usually do not consider individual details of smaller participant groups. A wide online cohort study that examined the participants of the Eureka Research Platform [28] showed that 476 out of 1480 responders continued to have PCC symptoms, and therefore, they were highly prevalent and common in their persistence during a median follow-up time of 360 days in a symptomatic group and 129 days in asymptomatic—the difference we could not notice because of the time of our limited time of observation. The authors found that the number of symptoms during acute infection, lower socioeconomic status, financial stress, and pre-COVID-19 depression were associated with PCC. In a recent populational study from South Africa [29], among participants infected during the Beta variant wave (the next major variant after the original Wuhan virus), 77.5% experienced ≥1 symptom at 1 month, while 64.3% experienced ≥1 symptom at 3 months, and 59.5% experienced ≥1 symptom at 6 months. As for the multivariable regression, factors associated with ≥1 persistent symptom at 6 months included age (40 and above), female sex, race (other than Black), the presence of any comorbidity (without any specific disease reaching statistical significance), the number of acute COVID-19 symptoms, COVID-19 severity, and SARS-CoV-2 variant (higher during the Beta wave than Omicron, which came much later). Interestingly, existing research usually points to the association of PCC with the female sex [30], while the pulmonary phenotype was present significantly more often among men in our study.

Because we did not manage to find any existing comparison between these post-COVID-19 phenotypes, we searched for research concerning disease courses that would adhere to our results; we assumed the lack of hospitalization necessity and oxygen administration to be the logical proof of little to no lung involvement. In a study among 147 patients infected with SARS-CoV-2 that were isolated without pharmacological interventions in a South Korean Center [31], 131 developed at least one of the following symptoms: fatigue, myalgia, memory impairment, hyposmia, hypogeusia, dizziness, and anxiety symptoms. Scientists revealed that cardiopulmonary symptoms improved over time but constitutional, neurological, and neuropsychiatric symptoms remained. Participants with remaining neuropsychiatric symptoms reported the lowest health-related quality of life (HRQoL), while factors associated with persistent symptoms and diminished HRQoL were identified as female sex, metabolic disease, and anxiety during the acute COVID-19 phase. The Norwegian cohort described similar symptoms among home-isolated convalescents, and they managed to associate them with increased convalescent antibody titers after 2 months, suggesting a more aggravated immune reaction to the virus [32]. These data also provide important information that symptoms persist despite being isolated in the place of living, which is a factor that is presumably crucial for the presentation of emotional symptoms.

When impaired exercise tolerance, dyspnea, and cough after COVID-19 are considered, we tend to be convinced that patients with preserved complaints would have lesions detectable with a CT scan. Indeed, our results suggest that there are importantly higher dyspnea scores among individuals with pulmonary involvement, and the majority of these patients with significant radiographic abnormalities will present alterations on PFTs. Interestingly, it does not mean that the group of patients without imaging abnormalities has no respiratory symptoms; there were previous studies that found no significant differences in PFTs when comparing patients with persistent COVID-19-related symptoms and asymptomatic ones, which suggests a much greater background than just respiratory system impairment [33]. Other symptoms that emerged in our study as more often associated with pulmonary involvement were myalgia and sleeplessness, both of which coexist with dyspnea [34,35] and bear the potential of aggravating patients’ poor self-assessment [30,36]. Premraj et al. associated these issues with hospital admission occurrence, which is consistent with our results [37].

European long-COVID recommendations [38], while concentrating on persistent lung impairment, suggest a careful evaluation of patients after COVID-19 and specify several factors suggesting the evidence of lung fibrosis, namely: single breath T_L,CO_ reduction, persistent ground glass opacities (GGOs) in radiological examinations, fatigue or anxiety 1–8 months after infection, number of symptoms during the first week of infection, older age, female sex, the severity of COVID-19, ARDS, high flow nasal oxygen treatment, mechanical ventilation and its length, involved lung parenchyma at discharge, length of hospital stay, elevated LDH and D-dimer level on admission, decreased lymphocyte T level, and prolonged Il-6 elevation. This guideline and the results presented in the present paper are consistent with our previously published data about the association of a longer hospital stay and oxygen administration with lower spontaneous SpO2 measured in patients’ home settings [39]. More common steroid use in the pulmonary phenotype group is just the consequence of early results from the RECOVERY trial proving reduced mortality among patients with COVID-19 and respiratory failure after dexamethasone use, which was the first unwavering pharmacological intervention in the early pandemic [40]. Participants with no lung involvement presented no respiratory failure and had no indications for steroid use at the time of the study.

We managed to connect COVID-19 lung involvement to existing hypertension and liver failure. While hypertension is one of the most common diseases in the general population, its coexistence with pulmonary COVID-19 has been vastly discussed because SARS-CoV-2 infiltration of human cells occurs via the angiotensin-converting enzyme 2 presented on the cell surface, and its expression is being modified with antihypertensive drugs [41,42]. The analysis of 1590 Chinese patients proved that hypertension is one of the risk factors of ICU admission, invasive ventilation or death during COVID-19, even after adjusting for age and smoking status [43]. Liver failure (especially liver cirrhosis) affects the immune system function, increases thrombotic and hemorrhagic risk, and impairs protein metabolism, making patients vulnerable and frail. In terms of COVID-19, it is expressed by an increased mortality and hospitalization rate, although existing data differ, which is probably because of different liver disease stages with the greatest correlation among patients with liver cirrhosis [44,45,46]. The meta-analysis of Middleton et al. [47] has shown liver cirrhosis to be linked with all-cause mortality in COVID-19. The smoking status itself was not significant, but the cumulated pack-years (*p* = 0.04) made the difference, plausibly impairing the lung volumes as an independent risk factor.

In a review of research concerning imaging abnormalities, the percentages were 35 to 72% in 1–6 month follow-up [48], while among those who underwent mechanical ventilation, it could reach even 96% at 3-month evaluation [49]. A meta-analysis that included 46 research papers [50] and assessed CT scans at 3–6 months after hospitalization revealed that the overall estimated proportion of chest CT inflammatory changes was 49% (95% CI 39–59%), and the overall estimated proportion of fibrotic changes was 34% (95% CI 25–43%).

Regardless of calculated percentages, modern methods of radiological analysis such as a contrastive learning model used by Li et al. [51] manage to distinguish clusters of post-COVID-19 patients—in the mentioned study, it was an air-trapping pattern (due to some small airways disease involvement) and an interstitial fibrotic-like pattern, while the latter were hospitalized more often (90 vs. 67%) and had lower FVC%, TL,CO%, TLC, and higher RV/TLC as well as more ground glass opacities during assessment after the median time of 113 days from COVID 19. These data look promising regarding its comparability to our clinical phenotype distinction; the pulmonary phenotype group presented with alterations in T_L,CO_%, RV%, TLC% and absolute values of FVC and FEV1. From the physiological point of view, Mancini et al. assessed 41 patients with cardiopulmonary exercise testing and discovered circulatory impairment and abnormal ventilatory patterns to be common in patients with PASC (with a component of dysfunctional breathing and resting hypocapnia), but even 46% met the criteria for myalgic encephalomyelitis/chronic fatigue syndrome [52].

Last but not least, we found several statistically significant differences between the study subgroups’ laboratory biomarkers: troponin, D-dimer, creatinine, WBC, monocyte and immature granulocyte count along with medium red cell volume and size (RDW-CV, RDW-SD). In a cross-sectional study, Maamar et al. found that a higher neutrophil count correlated with post-COVID-19 fatigue in men and post-COVID-19 anosmia in young women [53]. Townsend et al. evaluated both sexes after an acute COVID-19 and described no differences between the white blood cell count, neutrophils, lymphocytes, NL ratio, LDH or CRP, and post-COVID-19 fatigue (measured with the Chalder Fatigue Score CFQ-11) among 128 participants [54]. Severe fatigue-presenting patients were more often females and more often reported a previous history of anxiety/depression. On the other hand, 12% to 20% of hospitalized patients with COVID-19 (with predominantly affected lungs) present cardiac injuries with raised cardiac troponin levels [55], and the prothrombotic tendencies measured with the elevation of D-Dimer persist to 12 months after infection [56]. We did not manage to find any data on the importance of red cell distribution width, monocyte and immature granulocyte count after COVID-19. Evaluating the importance of differences found in our cohort requests further studies.

Finally, modern machine learning methods were also used to identify risk factors of delayed recovery at the 6-month follow-up [57]. Multimorbidity, malignancy, male sex, prolonged hospitalization, ICU stay and immunosuppressive therapy were significantly associated with persistent CT and PFTs changes. Results from laboratory test analyses remain inconsistent with our results, and algorithms only managed to associate D-dimer (>500 pg/mL) with lung function abnormalities. Persistently elevated inflammatory markers (IL-6 and CRP) were risk factors for unfavorable outcomes, while low serum anti-S1/S2 IgG and ambulatory acute COVID-19 correlated with better recovery [57].

At the time of conducting this research, we were the only pneumology outpatient clinic in the region that aggregated patients after COVID-19 and performed their wide routine laboratory, functional and radiological assessment. We believe our study attractively illustrates the natural history of SARS-CoV-2 infection, as none of the patients became vaccinated against the virus.

There are some limitations of the study which should be addressed. Participants did not strictly fulfill the post-COVID-19 condition definition during their first assessment (following World Health Organization: persisting symptoms over 12 weeks from the COVID-19 infection). Additionally, the observational study design could not consider confounding factors, especially in terms of the research group heterogeneity, the time of assessment from SARS-CoV-2 infection diagnosis and pharmacological therapy used during the acute phase (several patients received remdesivir, tocilizumab and fresh frozen plasma when hospitalized, respectively: 16, 6 and 20 individuals; there were no oral antivirals available at the time of the study). Our cohort was most likely infected with the original Wuhan strain, and the PCC course could not be the same for later strains such as Delta and Omicron.

## 5. Conclusions

While the origins of a specific organ involvement are not fully understood, we managed to distinguish patients with a pulmonary and a non-pulmonary PCC phenotype. Developing knowledge about the individualized nature of PCC could ultimately lead us to propose different therapeutic approaches. Considering the factors discussed so far, we intend to aid clinicians in making decisions concerning pulmonary specialists and pulmonary rehabilitation care on one hand (while lung and chest involvement is obvious) as well as proposing behavioral therapy on the other hand (when cognitive symptoms dominate). Nevertheless, our findings require confirmation in further studies, especially in terms of possible alternatives in the therapeutic approach and their outcomes.

## Figures and Tables

**Figure 1 biomedicines-11-02694-f001:**
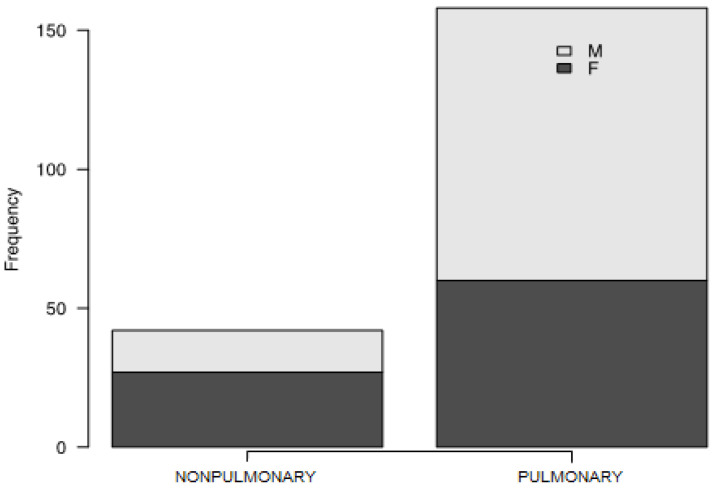
Frequency of non-pulmonary and pulmonary phenotype in the group studied.

**Figure 2 biomedicines-11-02694-f002:**
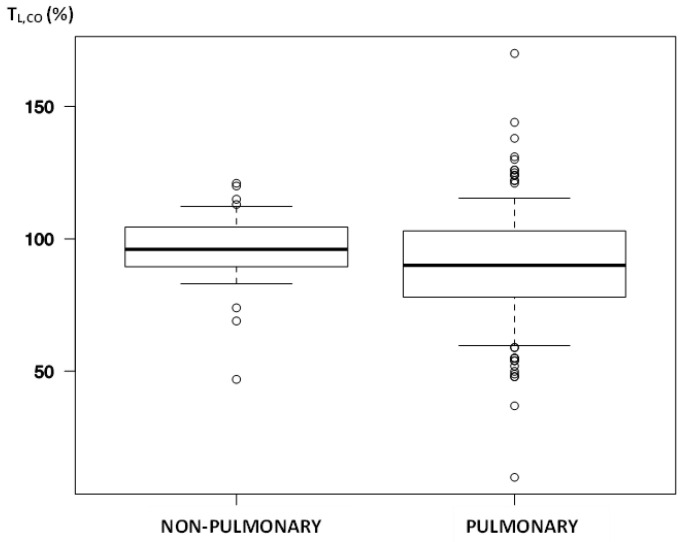
The median of lung transfer for carbon monoxide among participants after dividing for phenotype.

**Figure 3 biomedicines-11-02694-f003:**
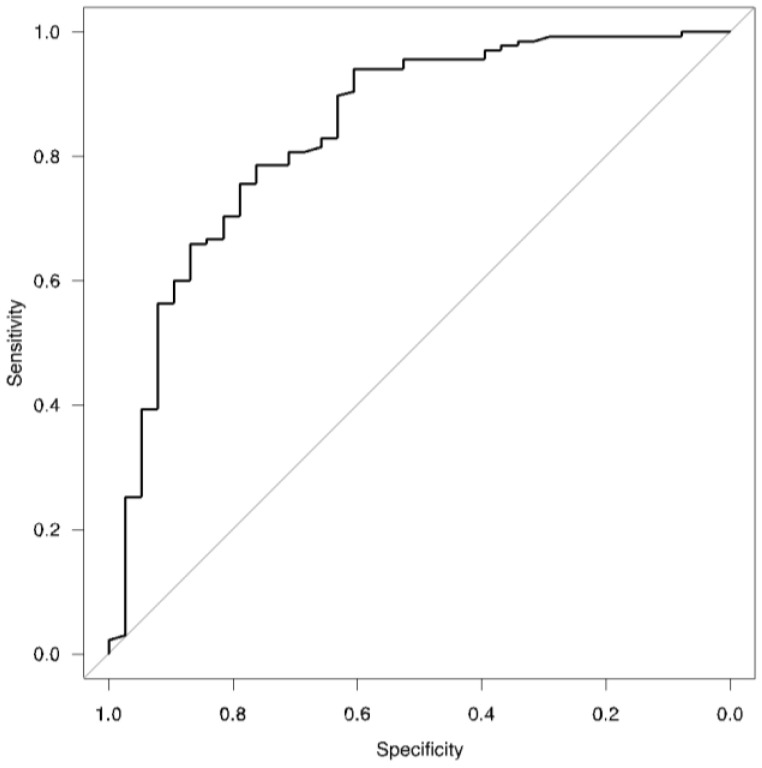
ROC for logistic regression model (Table 3).

**Table 1 biomedicines-11-02694-t001:** Characteristics of study participants. Abbreviations: BMI—body mass index; COPD—chronic obstructive pulmonary disease.

Parameter	Value
Age, years, median (IQR)	60 (50–67)
Males, *n* (%)	155 (54.96%)
BMI, kg/m2, median (IQR)	28.09 (25.61–31.65)
General health assessment, 0–100 numeric scale	70 (50–80)
Hypertension, *n* (%)	136 (48.23%)
Obesity, *n* (%)	69 (24.47%)
Diabetes, *n* (%)	52 (18.44%)
Asthma/COPD, *n* (%)	38 (13.48%)
Heart failure, *n* (%)	34 (12.06%)
Kidney failure, *n* (%)	6 (2.12%)
Liver failure, *n* (%)	7 (2.48%)
History of arterial thromboembolic events, *n* (%)	30 (10.64%; 14 myocardial infarctions; 16 stroke)
History of neoplasm, *n* (%)	21 (7.45%)
Obstructive sleep apnea, *n* (%)	1 (0.004%)
Hypothyroidism, *n* (%)	14 (4.96%)
Invasive ventilation, *n* (%)	2 (0.01)
Non-invasive ventilation, *n* (%)	2 (0.01)
Oxygen therapy, *n* (%)	114 (40.43%)
Corticosteroid, *n* (%)	107 (37.94%)
Remdesivir, *n* (%)	16 (5.67%)
Tocilizumab, *n* (%)	6 (2.12%)
Fresh frozen plasma, *n* (%)	20 (7.09%)
Antibiotic, *n* (%)	131 (46.45%)

**Table 2 biomedicines-11-02694-t002:** Comparison of the statistically important differences between the pulmonary and non-pulmonary participant groups.

Parameter	Mean (SD) or Median (IQR)	Pulmonary	Non-Pulmonary	*p*-Value
Males, %	56.5%	49%	7.5%	0.002
Age, years	57.26 (12.20)	58.87 (11.71)	51.21 (12.26)	<0.001
Pack-years	18 (10–25)	20 (10–30)	10 (7–20)	0.044
Spontaneous saturation, %	97 (95–98)	97 (95–98)	98 (97–99)	0.025
Dyspnea during acute phase, 1–5 numeric scale	3 (1–4)	3 (2–4.25)	2 (1–3)	0.000
Duration of dyspnea, days	12 (6–20)	14 (7–29)	10 (4–13)	0.023
Duration of myalgia, days	9 (5–14.25)	10 (6–15)	7 (5–10)	0.041
Dyspnea during assessment, 1–5 numeric scale	1 (0–2)	1 (0–2)	0 (0–1)	0.000
Hospitalization length, days	11 (7–16)	12 (7.75–17)	8.5 (6.75–11.25)	0.027
CT control score, points	1 (1–2)	2 (1–3)	0 (0–0)	0.000
FEV_1_, %	89 (79–98)	88 (78–96)	92.5 (82.75–99)	0.052
FVC, %	87 (78–97)	87 (77–97)	89.5 (84–101.25)	0.042
RV, %	99 (85–113)	96 (79–109.75)	111 (97.5–124)	0.000
T_L,CO_, %	92 (80–103)	90 (78.25–103)	96 (89.5–104.5)	0.022
TLC, %	97.79 (14.37)	95.91 (14.12)	104.44 (13.38)	<0.001
Cardiac troponin, ng/L	6 (3–9)	7 (4–12)	3.5 (3–5)	0.003
D-Dimer, ng/mL	358 (248–598)	365 (265.5–619)	286 (182–469.75)	0.004
WBCs, G/L	6.8 (6–8.025)	6.9 (6–8.1)	6.35 (5.65–7.225)	0.032
Immature cells, G/L	0.02 (0.01–0.04)	0.02 (0.01–0.04)	0.01 (0.01–0.02)	0.002
Monocytes, G/L	0.6 (0.5–0.7)	0.6 (0.5–0.7)	0.5 (0.4–0.6)	0.002
RDW-CV, %CV	13.4 (12.8–14.4)	13.4 (12.9–14.525)	12.9 (12.4–13.65)	0.004
RDW-SD, fl	43.9 (41.675–47.875)	44.9 (42.025–48.425)	42.8 (41.2–44.95)	0.008
Creatinine, mg/dL	0.82 (0.71–1)	0.86 (0.7375–1.0225)	0.75 (0.635–0.875)	0.004

Abbreviations: CT—computed tomography; FEV_1_—forced expiratory volume in 1 s; FVC—forced vital capacity; RDW-CV—red blood cell distribution width—coefficient of variation; RDW-SD—red blood cell distribution width—standard deviation; RV—residual volume; T_L,CO_—transfer factor of the lung for carbon monoxide; WBCs—white blood cells.

**Table 3 biomedicines-11-02694-t003:** Results of logistic regression—enhanced risk of pulmonary phenotype.

Coefficient	Estimate	OR	95% CI	*p*-Value
Intercept	−5.22	0.005	0.0005–0.07	<0.001
Age	0.07	1.07	1.03–1.11	<0.001
Male sex	1.75	5.73	2.23–14.7	<0.001
Low T_L,CO_	1.56	4.77	1.18–19.2	0.03
Severity of dyspnea during COVID-19	0.58	1.78	1.34–2.37	<0.001
AUROC	0.842	-	0.77–0.92	-

## Data Availability

Data available on reasonable request.

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
