# Peer review of "Is Pulmonary Involvement a Distinct Phenotype of Post-COVID-19?"

_biomedicines, 2023, doi:10.3390/biomedicines11102694_

Round 1

Reviewer 1 Report

GENERAL COMMENTS: This manuscript describes a cohort of post-COVID 19 with persistent symptoms which they have identified as having a post COVID condition (PCC) with a predominantly pulmonary phenotype.  They have identified some factors which identify this group of patients from prospectively obtained data, but their findings are mostly intuitive and expected based on patient presentation. Specifically, they have identified those with persistent pulmonary PCC as older middle-aged men, higher volume smokers (20 pack years) with more respiratory symptoms at onset of their illness, as more likely to have persistent abnormalities in respiratory symptoms and lung function studies, especially lung volumes and transfer factor (diffusing capacity).  The abnormalities noted are mostly in the normal range but lower than their comparison cohort who did not have pulmonary symptoms.  In other words, it seems that COVID patients with pulmonary symptoms are more likely to have persistent pulmonary PCC manifestations.  it is not entirely clear if these findings would impact medical management.  There are some other areas which may provide additional insight into this pulmonary PCC, which merit further comment.

 SPECIFIC COMMENTS:

Severity of illness:  No severity of illness measure was provided in this manuscript. However, the pulmonary PCC patients seemed to have a more severe illness than their cohorts.  While there may not have been a prospective severity of illness score, one may be able to be created based on the data they collected.  In conjunction with their other data, it may be possible to create a model that could predict the occurrence of pulmonary PCC.

CT control score/radiologic abnormalities: The authors comment that radiographic abnormalities were noted more frequently in the pulmonary PCC group, but do not otherwise elaborate on these radiographic abnormalities.  Specific areas of interest would be the types of radiographic abnormalities visualized which have been reported to range from ground glass opacities to fibrosis.  It would seem that these types of abnormalities could be correlated with the abnormalities in lung function specifically the transfer factor which in turn provide providers with information about the severity of lung involvement, need for further investigation and possibly even systemic treatment.  They also do not report the proportion of patients with the CT abnormalities which would also be important information.  The authors do mention these radiographic abnormalities noted by others in their discussion, but do not report on findings in their patient cohort.  Notably, their non-pulmonary PCC cohort had no CT score abnormalities. This information should be included in their manuscript.

 Co-variables: They also identified hypertension and liver failure as occurring more frequent in the pulmonary PCC group, but this may be more a manifestation of other variables including gender and age as opposed to COVID.  It is unclear if controlling for age and gender would identify these two variables as predictive of pulmonary PCC. There are also a host of other laboratory variables (D-dimer, creatinine, RDW, etc.), but as noted with the lung function studies, while different from non-pulmonary PCC, within the normal range.  The importance of these abnormalities without a prediction model is difficult to assess.

Outcomes:  Table 3 mentions poor outcomes in their pulmonary PCC patients, but it is not otherwise defined in their manuscript.

Reviewer 2 Report

This study invstigated the association between initial pulmonary involvement and the post-COVID-19 condition. Although it is interesting, I have several concerns.

1. Their findings may be too old to be applied in this Omicron wave.

2. In the result part of abstract section, please add some detailed data rather qualitive description.

3. Please define post-COVID condition in this study and cite associated references.

4. Please add the limitation about vaccination and oral anti-viral treatment.

Round 2

Reviewer 1 Report

GENERAL COMMENTS:  As previously noted, this manuscript in largely confirmatory in their findings that patents with a predominantly pulmonary post COVID condition (PCC) phenotype have more respiratory symptoms than their counterparts without pulmonary PCC manifestations. The CT scan probably predicts this involvement as they report CT control scores that are much different in those with pulmonary PCC than those who do not have pulmonary PCC. They need to further emphasize this finding in their manuscript, even if they were not able to further analyze the radiographic patterns.

SPECIFIC COMMENTS:

CT control score: the higher score in the pulmonary phenotype group is presented in the next to the last paragraph on page 4. The comparison is incomplete. They need to also include that the comparison CT control score was zero in the nonpulmonary cohort, and the range of data was also zero.  This is listed in the table 2 but is also important to include in the body of the manuscript. This suggests that CT control score abnormalities may predict a pulmonary PCC. Conversely, the absence of any abnormality on a control CT score (zero), would essentially have a negative predictive value of 100% for a pulmonary PCC (estimated). This type of calculation and analysis will further strengthen their manuscript.

Author Response

Thank you very much for taking a lot of time to thoroughly review our manuscript and for all the improvements it provided.

Response to your comment:

CT control score: the higher score in the pulmonary phenotype group is presented in the next to the last paragraph on page 4. The comparison is incomplete. They need to also include that the comparison CT control score was zero in the nonpulmonary cohort, and the range of data was also zero.  This is listed in the table 2 but is also important to include in the body of the manuscript. This suggests that CT control score abnormalities may predict a pulmonary PCC. Conversely, the absence of any abnormality on a control CT score (zero), would essentially have a negative predictive value of 100% for a pulmonary PCC (estimated). This type of calculation and analysis will further strengthen their manuscript.

We added the extended commentary to the body of the manuscript. It is true that persistent abnormalities on imaging appear as specific to the pulmonary group and it is logical together with other findings from the cohort studied. On the other hand, the disappearance of lesions was not observed in our group so we did not manage to calculate the negative predictive value based on this parameter.

Reviewer 2 Report

The authors response well, so I have no more comment.

Author Response

Once more, we are very grateful for your time and constructive review.